# Changes in Plasma Pyruvate and TCA Cycle Metabolites upon Increased Hepatic Fatty Acid Oxidation and Ketogenesis in Male Wistar Rats

**DOI:** 10.3390/ijms242115536

**Published:** 2023-10-24

**Authors:** Simon Nitter Dankel, Tine-Lise Kalleklev, Siri Lunde Tungland, Marit Hallvardsdotter Stafsnes, Per Bruheim, Thomas Aquinas Aloysius, Carine Lindquist, Jon Skorve, Ottar Kjell Nygård, Lise Madsen, Bodil Bjørndal, Magne Olav Sydnes, Rolf Kristian Berge

**Affiliations:** 1Department of Clinical Science, University of Bergen, N-5021 Bergen, Norwaythomas.aloysius@uib.no (T.A.A.); jon.skorve@uib.no (J.S.); ottar.nygard@uib.no (O.K.N.); bodil.bjorndal@hvl.no (B.B.); 2Department of Chemistry, Bioscience and Environmental Engineering, University of Stavanger, N-4021 Stavanger, Norwaymagne.o.sydnes@uis.no (M.O.S.); 3Department of Biotechnology and Food Science, NTNU Norwegian University of Science and Technology, N-7491 Trondheim, Norwayper.bruheim@ntnu.no (P.B.); 4Department of Heart Disease, Haukeland University Hospital, N-5021 Bergen, Norway; 5Department of Clinical Medicine, University of Bergen, N-5021 Bergen, Norway; lise.madsen@uib.no; 6Department of Sports, Food and Natural Sciences, Western Norway University of Applied Sciences, N-5020 Bergen, Norway

**Keywords:** mitochondria, liver, fatty acid oxidation, ketogenesis, biomarkers

## Abstract

Altered hepatic mitochondrial fatty acid β-oxidation and associated tricarboxylic acid (TCA) cycle activity contributes to lifestyle-related diseases, and circulating biomarkers reflecting these changes could have disease prognostic value. This study aimed to determine hepatic and systemic changes in TCA-cycle-related metabolites upon the selective pharmacologic enhancement of mitochondrial fatty acid β-oxidation in the liver, and to elucidate the mechanisms and potential markers of hepatic mitochondrial activity. Male Wistar rats were treated with 3-thia fatty acids (e.g., tetradecylthioacetic acid (TTA)), which target mitochondrial biogenesis, mitochondrial fatty acid β-oxidation, and ketogenesis predominantly in the liver. Hepatic and plasma concentrations of TCA cycle intermediates and anaplerotic substrates (LC-MS/MS), plasma ketones (colorimetric assay), and acylcarnitines (HPLC-MS/MS), along with associated TCA-cycle-related gene expression (qPCR) and enzyme activities, were determined. TTA-induced hepatic fatty acid β-oxidation resulted in an increased ratio of plasma ketone bodies/nonesterified fatty acid (NEFA), lower plasma malonyl-CoA levels, and a higher ratio of plasma acetylcarnitine/palmitoylcarnitine (C2/C16). These changes were associated with decreased hepatic and increased plasma pyruvate concentrations, and increased plasma concentrations of succinate, malate, and 2-hydroxyglutarate. Expression of several genes encoding TCA cycle enzymes and the malate–oxoglutarate carrier (*Slc25a11*), glutamate dehydrogenase (*Gdh*), and malic enzyme (*Mdh1* and *Mdh2*) were significantly increased. In conclusion, the induction of hepatic mitochondrial fatty acid β-oxidation by 3-thia fatty acids lowered hepatic pyruvate while increasing plasma pyruvate, as well as succinate, malate, and 2-hydroxyglutarate.

## 1. Introduction

Cells obtain the chemical energy stored in fatty acids (FAs) by the sequential degradation of the fatty acids, preferentially via β-oxidation in the peroxisomes (very-long-chain FAs) and mitochondria (long-, medium-, and short-chain FAs), or via α-oxidation (peroxisomes) or ω-oxidation (endoplasmic reticulum) when the carbon-3 of FAs has methyl- or other functional groups. Altered mitochondrial fatty acid β-oxidation in the liver influences whole-body energy homeostasis and may contribute to several metabolic abnormalities, including metabolic syndrome, nonalcoholic fatty liver disease, type 2 diabetes, and other metabolic diseases [1,2,3]. Acetyl-CoAs produced by mitochondrial fatty acid β-oxidation are metabolized in the TCA cycle, but are combined to form ketones rather than enter the TCA cycle when the hepatic availability of carbohydrate-derived pyruvate and oxaloacetate is low [4] (Figure 1). Thus, during glucose deprivation, ketones act as an efficient alternative energy source in the brain, and also in tissues such as the heart and kidneys under more normal metabolic conditions. Higher concentrations of ketones, either via endogenous production or therapeutic administration, may help prevent NAFLD and lipid-related diseases [5,6]. Moreover, the altered flux of TCA cycle intermediates and related metabolites is seen in several diseases [7]. For example, in insulin-resistant mice with blunted mitochondrial fatty acid β-oxidation, increased TCA flux was found to promote hepatic oxidative stress and inflammation [8]. TCA cycle metabolite concentrations might therefore potentially serve as indicators of hepatic mitochondrial fatty β-oxidation, also independent of ketosis, with possible utility for detecting metabolic disease development and progression.

TCA cycle flux and ketogenesis are tightly controlled processes in the systemic coordination of the energy metabolism, involving fundamental redox mechanisms where metabolic diseases are characterized by a shift towards more oxidation and the associated generation of reactive oxygen species (ROS) [9]. Coordinated cellular metabolic effects upon shifts in the redox state are largely mediated by the pyridine nucleotides NADH and NADPH, which can be reduced to NAD^+^ and NADP, respectively. A major purpose of the TCA cycle is to use the chemical energy in acetyl-CoA to obtain the reducing power of NADH, which can then be used in a number of biological processes. Following the irreversible synthesis of the ketone bodies acetoacetate and β-hydroxybutyrate (β-OHB) via the enzyme 3-hydroxy-3-methylglutaryl CoA synthase (HMG-CoA synthase) [10], acetoacetate and β-OHB are interconverted with the NADH and NAD^+^ as cofactors. Importantly, the cellular oxidation status of the NADH/NAD^+^ and NADPH/NADP^+^ redox couples are signaled extracellularly via equilibration with the lactate/pyruvate (L/P) and β-hydroxybutyrate/acetoacetate (β/A) redox couples, where L/P reflects the overall cytosolic redox state and β/A the overall mitochondrial redox state [9]. Circulating ratios of these redox components can therefore inform cellular energetic states, such as increased hepatic lipid synthesis/storage vs. catabolism/expenditure and lowered glucose production in oxidized conditions, with implications for metabolic diseases [11].

Mechanistically, acetyl-CoA derived from pyruvate or mitochondrial fatty acid β-oxidation is an allosteric activator of pyruvate carboxylase (PC) in the liver, which converts pyruvate into oxaloacetate and thereby prevents ketogenesis by maintaining sufficient oxaloacetate in the TCA cycle [12]. At the same time, however, acetyl-CoA (along with NADH) is a competitive inhibitor of the enzyme complex pyruvate dehydrogenase (PDH), which converts glycolysis-derived pyruvate into acetyl-CoA, CO_2_, and NADH. This acetyl-CoA-mediated inhibition of PDH may support fatty acid oxidation and ketogenesis while sparing glucose (such as during fasting). Moreover, malonyl-CoA derived from the carboxylation of acetyl-CoA by acetyl-CoA carboxylase (ACC) serves as a critical metabolic switch by acting both as an allosteric inhibitor of carnitine palmitoyltransferase I (CPT1) (thereby inhibiting mitochondrial fatty acid β-oxidation) and as the initial component of the de novo synthesis of fatty acids [13,14].

In the present study in male Wistar rats, we sought to gain new fundamental insight into changes in TCA-cycle-related metabolite profiles during the hepatic prioritization of fatty acid over glucose oxidation. Specifically, our objective was to determine whether enhanced hepatic mitochondrial fatty acid β-oxidation is reflected in altered cellular and plasma concentrations, not only of ketones, but also intermediary substrates (e.g., pyruvate, lactate) and TCA cycle metabolites. To address this objective, we administered rats with synthetic fatty acids, known as 3-thia fatty acids, which activate mitochondrial biogenesis and the capacity of mitochondrial fatty acid β-oxidation in the liver, and which improve whole-body metabolic homeostasis in part by decreasing hepatic and plasma triglyceride levels [15]. These structurally modified fatty acids contain a sulfur atom rather than a methylene group in the carbon chain. Tetradecylthioacetic acid (TTA), where the 3-methylene group is substituted, is non-β-oxidizable, but can undergo ω-oxidation, while 2-(tridec-12-yn-1-ylthio)acetic acid (C_15_H_26_O_2_S, known as 1-triple TTA) (Figure 2), having a triple bond at the ω-1 position in addition to the sulfur atom, is resistant to both β-oxidation and ω-oxidation. 

Our data show that 3-thia fatty acid administration profoundly affects the pyruvate metabolism, with a decrease in hepatic pyruvate and an increase in plasma pyruvate concentrations. Accordingly, plasma pyruvate concentrations showed strong positive correlations with mitochondrial fatty acid β-oxidation, the activity of PDH, and concentrations of NAD^+^ and NADH in the liver. Moreover, the increased hepatic fatty acid oxidation was reflected in increased plasma concentrations of the TCA metabolites succinate, malate, and 2-hydroxyglutarate.

## 2. Results

### 2.1. The 3-Thia Fatty-Acid-Induced Alterations in Fatty Acid Oxidation

To study the metabolic changes resulting from increased fatty acid oxidation, we administered the 3-thia fatty acid analog 1-triple TTA to rats for three weeks. Compared to the control, the hepatic palmitoyl-CoA oxidation capacity was higher in rats treated with 1-triple TTA in the presence of the fatty acid oxidation inhibitor malonyl-CoA (Figure 3A). While malonyl-CoA inhibited fatty acid oxidation by 73% in the controls, the inhibitory effect of malonyl-CoA was decreased to 29% by 1-triple TTA administration (Figure 3B). The decreased malonyl-CoA sensitivity by this 3-thia fatty acid was accompanied by increased hepatic malonyl-CoA decarboxylase activity (Figure 3C) and decreased plasma malonyl-CoA levels (Figure 3D). These data show the potency of 1-triple TTA to increase hepatic fatty acid oxidation even in the presence of malonyl-CoA, and might be reflected in acylcarnitine concentrations. The plasma concentrations of palmitoylcarnitine (C16) were decreased to a greater extent (50% reduction) than acetylcarnitine (C2) in 1-triple TTA-treated rats compared to the controls, reflected by a higher C2/C16 ratio (Figure 3E–G), supporting a relative increase in the generation of the C2 (acetyl-CoA) end-products through mitochondrial β-oxidation of the C16 (palmitic) fatty acids.

Along with the increased fatty acid oxidation, the 1-triple TTA administration increased plasma concentrations of the ketone bodies β-hydroxybutyrate (β-OHB) and acetoacetate, and increased the ratio of total plasma ketone bodies over nonesterified fatty acids (NEFAs) (Figure 3H–J). The increased hepatic mitochondrial fatty acid (palmitoyl-CoA) oxidation, furthermore, corresponded to the increased gene expression of *Cpt2* (encoding carnitine palmitoyltransferase (CPT)-2, involved in fatty acid transport into the mitochondria) and *Hmgcs2* (3-hydroxy-3-methylglutaryl-CoA synthase 2 (a mitochondrial enzyme acting at the first step of ketogenesis)) (Figure 3K), in turn corresponding to strongly increased CPT2 and HMG-CoA synthase activities (Figure 3L). Across the control and treatment groups, correlation analysis showed strong correlations between several of these hepatic measures of mitochondrial fatty acid oxidation and plasma malonyl-CoA, as well as C2 and C16 fatty acid concentrations (inverse correlations), and the plasma C2/C16 ratio and ketone concentrations (positive correlations) (Figure 3M). Hepatic concentrations of NAD^+^ and NADH also closely followed these patterns (Figure 3M).

### 2.2. Relationship between Hepatic Fatty Acid Oxidation and Circulating Pyruvate, Lactate, and Ketones

We next sought to determine whether the circulating concentrations of pyruvate and lactate reflected the increased hepatic mitochondrial fatty acid oxidation in rats treated with 3-thia fatty acids. The 1-triple TTA administration to rats lowered the hepatic concentration of pyruvate, while the lactate concentration remained constant in the liver and increased in plasma (Figure 4A,B). Long-term administration of TTA for 50 weeks provided a similar pattern as 3-week 1-triple TTA in the liver and plasma concentrations of lactate and pyruvate, although with large interindividual variation and not significant (Figure 4B). Conversely, the plasma concentration of pyruvate was increased by 1-triple TTA administration (Figure 4A). This opposite effect on pyruvate concentrations in the liver and plasma was associated with a marked decrease in the activity of hepatic PDH activity, which converts pyruvate into acetyl-CoA for entry into the TCA cycle (Figure 4C). Accordingly, the plasma pyruvate concentrations showed a strong inverse correlation with hepatic pyruvate concentrations (Figure 4D). Concomitantly, plasma pyruvate showed strong positive correlations with hepatic malonyl-CoA-sensitive fatty acid β-oxidation, PDH activity, and NAD^+^ and NADH concentrations (Figure 4D). Plasma pyruvate concentrations, furthermore, showed an inverse correlation with plasma malonyl-CoA concentrations, and strong positive correlations with ketones and the C2/C16 acylcarnitine ratio (Figure 4E). Conversely, within the liver, the hepatic malonyl-CoA-sensitive hepatic fatty acid β-oxidation showed a strong inverse correlation with hepatic pyruvate (Spearman’s rho = −0.780) and strong positive correlations with hepatic NAD^+^ (rho = 0.792) and NADH (rho = 0.701) (all *p* < 0.001). Taken together, plasma pyruvate positively reflected TTA-induced fatty acid β-oxidation, and was associated with a decreased hepatic pyruvate flux into the TCA cycle.

Pyruvate forms a redox couple with lactate that reflects the overall cytosolic redox state, and this couple plays an important role in the energy metabolism in equilibrium with other redox couples, including NAD^+^/NADH and β-OHB/acetoacetate, which reflect mitochondrial redox state [9]. The 1-triple TTA administration increased the hepatic lactate/pyruvate ratio (Figure 4F), largely due to decreased hepatic pyruvate (Figure 4A), and this was mirrored by a lowered plasma lactate/pyruvate ratio (Figure 4F). Thus, the plasma lactate/pyruvate ratio was significantly inversely correlated with the hepatic lactate/pyruvate ratio across the control and 1-triple TTA-treated rats (Figure 4G). Although the treatment had no clear impact on the hepatic NAD^+^/NADH ratio or the plasma β-OHB/acetoacetate ratio (Figure 4F), the plasma lactate/pyruvate ratio correlated strongly and positively with the plasma β-OHB/acetoacetate ratio (but not with the NAD^+^/NADH ratio) (Figure 4G). Additionally, the plasma lactate/pyruvate ratio correlated inversely with the hepatic malonyl-CoA-sensitive mitochondrial β-oxidation (rho = −0.629, *p* = 0.001). Taken together, these data connect increased hepatic mitochondrial fatty acid β-oxidation particularly to a reduced plasma lactate/pyruvate ratio.

### 2.3. Induction of Hepatic Fatty Acid Oxidation Alters Liver and Plasma Levels of TCA Cycle Intermediates and the Expression of Related Genes

Because the oxidation of acetyl-CoA to CO_2_ by the TCA cycle is a central process in the energy metabolism, and higher TCA cycle intermediate concentrations may reflect an increased TCA cycle flux, we next evaluated the effect of 1-triple TTA administration on TCA cycle intermediates in the liver and plasma. In the liver, the treatment had no significant effects on the measured intermediates, except for hepatic 2-hydroxyglutarate, a conversion product from α-ketoglutarate, which increased 4.6-fold (Table 1). The treatment also increased plasma 2-hydroxyglutarate concentrations by 50% (Table 1). Additionally, 1-triple TTA increased the plasma concentrations of succinate and malate by 180% and 60%, respectively (Table 1).

The 1-triple TTA treatment significantly and robustly increased the hepatic mRNA expression of several genes involved in the TCA cycle, including citrate synthase (*Cs*), aconitase 2 (*Aco2*), and isocitrate dehydrogenase (*Idh1*) (Figure 5). TTA administration, however, did not significantly alter the hepatic mRNA expression of *Idh2* (isocitrate dehydrogenase), which encodes the mitochondrial as opposed to the cytosolic isoform of the IDH enzyme that converts isocitrate to 2-oxoglutarate (Figure 5). Furthermore, the hepatic gene expression of *Dld*, *Ogdh*, and *Dlst* was increased in 1-triple TTA-treated rats compared to the controls (Figure 5). These genes encode the subunits that comprise 2-oxoglutarate dehydrogenase (OGDC), the enzyme that catalyzes the conversion of 2-oxoglutarate (derived from 2-hydroxyglutarate) to succinyl-CoA. Moreover, along with the increased plasma succinate concentrations (Table 1), mRNA for the ATP-specific succinyl-CoA synthase (SUCLA2) was also increased after the 3-thia fatty acid administration (Figure 5). Also, the gene encoding succinate dehydrogenase (SDHA), which catalyzes the interconversion of succinate to fumarate, was significantly upregulated by 1-triple TTA administration (Figure 5). Finally, the increased plasma malate concentrations after administration of 3-thia fatty acid (Table 1) corresponded to increased hepatic gene expression of fumarase (*Fh*) and malate dehydrogenase 1 and 2 (*Mdh1*, *Mdh2*) (Figure 5). Together, these data reveal a strong upregulation of several genes with rate-limiting functions in the TCA cycle, and with a change in plasma concentrations of specific TCA cycle intermediates, e.g., 2-oxyglutarate, succinate, and malate (Figure 6). 

### 2.4. Effects on the Mitochondrial Solute Carrier (SLC) 25 Transporters

TCA cycle intermediates, which are hydrophilic and charged molecules, cannot readily diffuse across membranes, but rely on channels, pumps, and/or transporters to move in and out of cells and subcellular organelles. Therefore, we also examined how 1-triple TTA affected SLC25-family proteins, which shuttle a variety of metabolites across the mitochondrial inner membrane. The hepatic gene expression of *Slc25a10*, which transports dicarboxylic acids, such as malate and succinate, remained unaltered by 1-triple TTA administration (Figure 5). Similarly, *Slc25a12,* facilitating the transport of cytoplasmic glutamate with mitochondrial aspartate, was unchanged. However, the 3-thia fatty acid treatment increased the hepatic gene expression of *Slc25a11* (Figure 5). This carrier transports 2-oxoglutarate across the inner membrane of mitochondria in an electroneutral exchange for malate and other dicarboxylic acids, such as aspartate and isocitrate. On the other hand, hepatic expression of *Slc25a13* was significantly lowered by 1-triple TTA administration (Figure 5). This protein catalyzes the exchange of aspartate for glutamate and a proton across the inner mitochondrial membrane.

### 2.5. The 3-Thia Fatty Acid Administration Alters TCA-Related Enzyme Activity and Expression

To gain further insight into the mechanisms governing the 3-thia fatty-acid-induced alterations in TCA-related metabolites, we finally examined how 1-triple TTA administration affected the activity of anaplerotic and cataplerotic enzymes. Interestingly, the major anaplerotic enzyme pyruvate carboxylase (*Pc*), which generates oxaloacetate directly to the mitochondria from pyruvate, did not show altered hepatic gene expression after 1-triple TTA administration [16]. Moreover, we reported that genes encoding other cataplerotic enzymes were unaffected (ATP citrate (*Acly*) lyase and aspartate aminotransferase (*Ast*)) or slightly downregulated (*Pepck*) in the liver of rats treated with 3-thia fatty acids [16]. We here also report unchanged gene expression of coactivator-associated arginine methyltransferase 1 (*Carm1*), considered as an epigenetic modulator (Figure 5). On the other hand, the hepatic gene expression of glutamate dehydrogenase (*Gdh*) was significantly increased by 1-triple TTA, suggesting the increased conversion of glutamate into the TCA intermediate 2-oxoglutarate (also known as α-ketoglutarate) (Figure 5). The plasma activities of alanine transaminase (ALAT), aspartate transaminase (ASAT), creatine kinase (CK), and plasma carbamide (urea) were not altered by 1-triple administration (Figure 7).

## 3. Discussion

This study demonstrates that mitochondria-targeted 3-thia fatty acids, previously shown to stimulate hepatic mitochondrial biogenesis and hepatic fatty acid oxidation capacity even in the presence of malonyl-CoA [17], increased the plasma concentrations of pyruvate and the TCA cycle metabolites 2-hydroxyglutarate, succinate, and malate in vivo in male Wistar rats. These changes were, furthermore, reflected in increased plasma pyruvate, as well as increased plasma ketone levels, and ratios of C2/C16 acylcarnitine and ketones/NEFA. In the liver, the 3-thia fatty acid treatment upregulated the expression of all measured genes encoding TCA cycle enzymes, along with the malate–oxoglutarate carrier (*Slc25a11*) and glutamate dehydrogenase (*Gdh*).

Our data suggest that plasma pyruvate concentrations may, to a large extent, reflect hepatic mitochondrial fatty acid β-oxidation and TCA cycle activity, at least as observed in male Wistar rats. The increased plasma pyruvate corresponded to lowered hepatic pyruvate concentrations, indicating increased hepatic disposal of pyruvate during enhanced mitochondrial β-oxidation. During fasting and low-carbohydrate availability, there is limited formation of pyruvate from glucose, in turn limiting oxaloacetate concentrations and TCA flux, and thereby promoting ketogenesis from accumulated acetyl-CoAs. Although pyruvate can be converted into oxaloacetate, and thereby glucose in gluconeogenesis, increased flux from pyruvate to oxaloacetate, however, does not appear to explain the decreased hepatic pyruvate concentrations. Firstly, we observed no significant change in hepatic oxaloacetate concentrations. Secondly, we previously reported that 3-thia fatty acid administration to male Wistar rats (a) did not affect the hepatic expression of *Pc* (the main anaplerotic enzyme by converting mitochondrial pyruvate into oxaloacetate [18]) and (b) lowered rather than increased the expression of *Pepck* (the rate-limiting gluconeogenic enzyme), concomitant with lowered hepatic glycogen stores and plasma glucose [16]. Notably, the 3-thia fatty acids lowered hepatic pyruvate concentrations despite also lowering the activity of PDH; this lowered PDH activity could be expected to limit the conversion of pyruvate into acetyl-CoA. Thus, the lowered hepatic pyruvate concentrations may have resulted from increased pyruvate disposal out of the hepatocytes, which, together with the lowered PDH activity, might also have contributed to the increased plasma pyruvate concentrations. On the other hand, increased plasma pyruvate might also have resulted from decreased terminal oxidation of pyruvate to CO_2_ in peripheral tissues, as observed in altered physiological states, such as insulin resistance and obesity [19].

We found no or minimal changes in hepatic and circulating lactate concentrations upon the 3-thia fatty acid administration. Lactate and pyruvate are both released by different metabolic organs, most abundantly by skeletal muscles, but also by other organs, including the liver and adipocytes [9,20]. However, the present study suggests that the liver primarily affects circulating pyruvate and not lactate in conditions of markedly enhanced hepatic mitochondrial fatty acid β-oxidation. Notably, previous studies have shown that the increased formation of pyruvate in favor of lactate is metabolically favorable. By corollary, supplementation of pyruvate was found to improve metabolic regulation in people with overweight, including decreasing body weight and fat mass, and promoting tolerance to physical exercise [21,22,23,24,25]. Previous studies have also highlighted the potential benefits of increased circulating pyruvate for neurodegenerative diseases, attributed at least in part to neuronal reductions in mitochondrial ROS levels and related cell damage and death [26,27]. As a redox partner with lactate, pyruvate acts as a potent antioxidant and anti-inflammatory agent by shifting the ratio of NADH/NAD^+^, a second redox couple whose circulating state is critical for intracellular redox balance across the body [9]. Consistently, direct manipulation of the extracellular lactate/pyruvate ratio towards pyruvate has been shown to normalize (decrease) the intracellular NADH/NAD^+^ ratio and thereby mitigate the reductive stress that is implicated in multiple diseases (e.g., allowing appropriate cellular respiration and proliferation) [28]. Of note, in the present study, we measured pyruvate and lactate in whole-liver tissue and did not distinguish between the mitochondrial and cytosolic fractions. This could be relevant because pyruvate increases NADH (reductive stress) in the mitochondria while acting as an oxidizing agent in the cytosol [9]. In line with this, we recently found that TTA treatment upregulated NAD^+^ synthesis and further stimulated the conversion of tryptophan to nicotinamide, a precursor for NAD^+^, thus providing the NAD^+^ necessary for continued fatty acid β-oxidation [15,17].

Interestingly, our data indicate that the 3-thia fatty acids induce ketogenesis, even on a high-carbohydrate diet, and while increasing TCA cycle activity, as measured by upregulated genes encoding hepatic TCA cycle enzymes and increased plasma TCA metabolites. For example, there was increased mRNA expression for citrate synthase (Cs), the enzyme that combines oxaloacetate and acetyl-CoA into citrate in the first step of the TCA cycle. In the presence of sufficient oxaloacetate, this increase in *Cs* expression is expected to direct acetyl-CoA moieties away from mitochondrial ketogenesis and towards lipogenesis in the cytosol; lower *Cs* expression, conversely, is expected to increase ketogenesis by promoting the accumulation of acetyl-CoA. However, acetyl-CoA for ketogenesis can also accumulate directly from β-oxidation during increased fatty acid availability [29], which may come from adipose tissue in a catabolic state [30]. It should be noted that we did not distinguish between mitochondrial and cytosolic oxaloacetate in the liver. Nonetheless, upon the 3-thia fatty acid administration, we observed increased malate concentrations in plasma, which may reflect a shift in the oxaloacetate/malate balance towards malate, which is known to occur during increased mitochondrial fatty acid oxidation, and consequent increases in NADH and ATP synthesis. Altogether, the increased plasma malate, as well as succinate and 2-hydroxyglutarate (the latter derived from α-ketoglutarate), along with the upregulation of several TCA-driving genes, points to increased overall TCA cycle activity, but concomitant with specific changes in enzymatic activities towards the accumulation of acetyl-CoA and its conversion into ketone bodies, most notably HMG-CoA synthase and CPT2. HMG-CoA synthase is rate-limiting for ketogenesis [10], while TCA substrate flux and ketogenesis are influenced in part on CPT2, which transports fatty acids into the mitochondria for β-oxidation that yields acetyl-CoA moieties. The free CoA released from acetyl-CoA during ketogenesis, in turn, allows continued fatty acid β-oxidation.

Recent studies have implicated the release of TCA intermediates as a component of cellular signaling [7], and the "non-metabolic" functions of succinate, 2-hydroxyglutarate, fumarate, and acetyl-CoA have been found to modulate immunity and/or tumorigenesis, driving alterations of specific genes and the epigenome [31]. To communicate mitochondrial status, succinate generated in the mitochondrial matrix is exported to the cytosol via the SLC25A10 carrier through an exchange of succinate with malate, and SLC25A10 thus provides malate for the transport of citrate, which is required for fatty acid synthesis [32]. However, in the present study, we found no change in the expression of hepatic *Slc25a10* by TTA administration. On the other hand, our findings indicate a global enhancement of the cellular energy metabolism upon the 3-thia fatty acid treatment, including a significant upregulation of several genes encoding proteins important in the TCA cycle and related substrate fluxes through the malate–aspartate shuttle (e.g., *Slc25a11*, *Got2*, *Mdh1*, and *Mdh2*). Additionally, the treatment downregulated *Slc25a13*, a subunit of the mitochondrial amino acid carrier AGC2, responsible for transport of glutamate in exchange for aspartate. As glucose stimulates AGC2 activity, this downregulation is consistent with the 3-thia fatty-acid-induced hepatic fatty acid β-oxidation, and our previous observations that the 3-thia fatty acid administration to rats lowers the hepatic content of glycogen as well as glucose and plasma insulin levels [16]. Furthermore, the treatment increased the expression of *Gdh*, encoding an enzyme that catalyzes the conversion of L-glutamate to α-ketoglutarate using NAD^+^ or NADPH^+^ as a coenzyme. This enzyme can fuel the TCA cycle and thereby act as a link between catabolic and anabolic pathways, depending on the energy state of the body. GDH activity is elevated upon caloric restriction or low blood glucose, in line with the increase upon fatty acid oxidation after 3-thia fatty acid administration. Elevated flux via α-ketoglutarate due to increased IDH1 and GDH activity upon 3-thia fatty acid administration might explain the increased hepatic and plasma concentrations of 2-hydroxyglutarate (Figure 6). The 2-hydroxyglutarate has been found to modulate core metabolic processes, such as by increasing mTOR signaling downstream in the PI3K/AKT/TSC1-2 pathway [33]. Finally, we observed an upregulation of genes encoding the malic enzyme (malate dehydrogenase, *Mdh1*, and *Mdh2*) after 3-thia fatty acid administration, an important cataplerotic and lipogenic enzyme that catalyzes the oxidative decarboxylation of L-malate to pyruvate and CO_2_, with the concomitant reduction of the cofactor NAD^+^ or NADP^+^. The regulation of the malic enzyme is dependent on ATP as an inhibitor and fumarate as an activator [34], and recent studies have reported that the malic enzyme exhibits antioxidant properties and reduces ROS production by increasing the NADPH levels required for ROS scavenging [35].

Although smaller quantities of ROS play a key role in cell signaling and the initiation of core biological processes [36], and are essential in the physiological adaptation to environmental and other stressors [37], high levels of ROS are linked to oxidative stress associated with several pathologies, including metabolic syndrome, involving lipid, protein, and DNA damage, cell death, tissue injury, and more [38]. By promoting a more reduced redox state, the 3-thia fatty acid administration might enhance hepatic mitochondrial fatty acid β-oxidation and systemic energy homeostasis, while at the same time limiting the potential for oxidative stress associated with high mitochondrial activity and reduced ROS-scavenging capacity. In line with this, we have previously reported that TTA treatment protects against fish-oil-induced oxidative stress, reflected by hepatic/mitochondrial biomarkers of protein oxidation and lipoxidation [39]. Moreover, the interconversion of the lactate/pyruvate and β-OHB/acetoacetate redox couples depends in large part on the NAD^+^/NADH redox couple, which reflects the mitochondrial redox state [9]. Accumulation of NADH can potentially be toxic, and this effect is normally counteracted through a feedback loop where NADH inhibits the glucose, glutamine, and fatty acid oxidation pathways, which are the primary sources of NADH. Although we did not measure redox components in mitochondria and cytosol separately, the lowered hepatic pyruvate concentrations by the 3-thia fatty acid administration may have limited the accumulation of mitochondrial NADH, as more pyruvate in mitochondria increases NADH [9].

The mechanisms mediating the TTA-induced pyruvate decrease in the liver and increase in plasma, and the observed changes in other plasma metabolites might involve the activation of peroxisome-proliferator-activated receptor alpha (PPARα), an established master transcriptional regulator of hepatic fatty acid oxidation and mitochondrial proliferation. In vitro experiments in rat primary hepatocytes and the human hepatocyte cell line HepG2, as well as in vivo evidence of increased hepatic PPARα target gene expression, support that TTA acts as a pan-PPARα agonist [40,41]. Indeed, TTA was found to increase the hepatic expression of PPARα target genes in wild-type but not PPAR-α knockout mice [42]. As a relevant example of metabolite regulation, both PPARα [43] and TTA [16] robustly activate the expression of the malic enzyme, which converts malate into pyruvate in the cytosol, which could be expected to decrease general malate abundance and increase hepatic pyruvate. However, because TTA instead decreased the pyruvate concentration in the liver, the high generation of pyruvate from malate via the malic enzyme might have caused more pyruvate to leave the liver, potentially explaining the increased plasma pyruvate, and possibly also the decreased plasma malate, upon TTA treatment. Consistently, mice with a knockout of PPARα showed altered hepatic utilization of lactate/pyruvate pools for glucose production in feeding and fasting [44]. Notably, the effects of TTA on the hepatic expression of PPARα, along with altered plasma concentrations of metabolite markers (e.g., B-vitamins and related “one-carbon” metabolites), were found to be sustained in a long-term (50-week) experiment in male Wistar rats, indicating a lasting metabolic shift upon TTA administration [45], although the present study only found a trend of the altered pyruvate in the liver and plasma after 50 weeks, as seen after 3 weeks. The 50-week experiment also showed increased expression of fatty acid oxidation genes in the heart, although weaker than in the liver [46]. TTA-induced changes are, moreover, associated with cardiometabolic protection involving antioxidant effects and reduced inflammation, as exemplified in a relatively long-term (12-week) experiment in apoE knockout mice [47].

The present study has several limitations. Firstly, the analyses represent a snapshot of metabolic status, which could miss more dynamic changes. Secondly, future analyses of enzyme activities and TCA flux should help clarify the pathways of the observed pyruvate disposal in the liver, and to what extent the altered metabolism in peripheral tissues contributed to the observed changes in plasma pyruvate and other metabolites. For example, the use of stable isotope tracers would allow tracing the fate of specific metabolic substrates (e.g., of glucose, fatty acids, and pyruvate) to more precisely understand metabolite fluxes and the origins of plasma and tissue metabolite changes [48,49]. Also, the impact of pharmacologic enzyme inhibition on liver and plasma metabolites could be investigated, and redox states and metabolites could be measured in other tissues that may contribute to plasma metabolite concentrations, such as the skeletal muscle, adipose tissue, and the heart. Thirdly, although 3-thia FAs largely target mitochondrial biogenesis and fatty acid β-oxidation in the liver, we cannot rule out some direct effects in other tissues. Finally, the generalizability of the findings beyond male Wistar rats needs to be determined. Previous studies have demonstrated similar effects of 3-thia FAs on hepatic fatty acid oxidation in mice [50] and hamsters [51], and a lipid-lowering effect in male patients with type 2 diabetes [41], but TCA-related metabolite profiles upon a targeted increase in hepatic β-oxidation have not been measured in other species. Additionally, studies in females are lacking and might have shown different responses than in males.

In conclusion, we found that pharmacologic activation of hepatic mitochondrial fatty acid β-oxidation in male Wistar rats lowered hepatic pyruvate concentrations and increased plasma concentrations of pyruvate, succinate, malate, and 2-hydroxyglutarate. These changes corresponded to increased hepatic ketogenesis, and a broad upregulation of several genes involved in the TCA cycle and related metabolite fluxes. Our study has identified the potential plasma markers of these metabolic changes associated with enhanced fatty acid β-oxidation in the liver, whose clinical relevance should be further evaluated in humans. For example, increased serum pyruvate was recently reported as a prognostic marker for the early detection of patients at risk of critical COVID-19 [52], and our data suggest that this risk might involve altered energy metabolism in the liver.

## 4. Materials and Methods

### 4.1. Animal Study

The animal studies were conducted according to the Guidelines for the Care and Use of Experimental Animals, in accordance with the Norwegian legislation and regulations governing experiments using live animals, and approved by the Norwegian State Board of Biological Experiments with Living Animals (Permit numbers 2015-7367 [17] and 2005-140 [39]). Male Wistar rats, *Rattus Norvegicus*, 5 weeks old, were purchased from Taconic (Ejby, Denmark). Upon arrival, the rats were randomized using Research Randomizer, labeled, and placed in open cages, four in each cage, where they were allowed to acclimatize to their surroundings for one week. During the acclimatization and experiment period, the rats had unrestricted access to chow and tap water, and were kept in a 12 h light/dark cycle at a constant temperature (22 ± 2 °C) with a relative humidity of 55% (±5%). Upon the start of the experiments, the rats were block-randomized to their respective interventions, providing 8 rats per group. Due to the potent metabolic effects of 3-thia FAs, previous experiments found marked differences in metabolic phenotypes with as few as 3 rats per group [53], supporting the robust power in the present experiment. During the present 3-week-long experiment with 1-triple TTA [17], there were two rats in each cage separated with a divider that allowed them to have sniffing contact. All rats were provided chow throughout the experiment. In addition, the control group (*n* = 8) received 0.5 mL daily of 0.5% methylcellulose, and the intervention group (*n* = 8) received 100 mg/kg body weight 1-triple TTA (C_15_H_26_O_2_S, obtained from Synthetica AS, Oslo, Norway) dissolved in 0.5 mL 0.5% methylcellulose daily. Methylcellulose was provided orally by gavage by a technician blinded to the experimental setup. All animals were weighed daily and feed intake was determined weekly. In another rat experiment with TTA [39], the 10-week-old male Wistar rats were housed five per cage and had free access to water and food during the 50 weeks of study. The control group (*n* = 9) was fed a high-fat diet with 25.5% (*w*/*w*) fat, consisting of 23.5% lard and 2% soybean oil. The TTA group (*n* = 9) was fed a high-fat diet supplemented with TTA (0.375% (*w*/*w*) of the feed, approximately 300 mg/day/kg body weight dependent on total food intake).

At sacrifice, rats, equally divided between groups throughout the morning, were anaesthetized by inhalation of 2–5% isoflurane (Schering-Plough, Kent, UK); the abdomen was opened along the midline, and exsanguination was performed using cardiac puncture. EDTA blood was collected and immediately chilled on ice. The samples were centrifuged, and plasma was stored at −80 °C prior to analysis. The liver was collected and weighed, and a fresh sample from each liver was used for β-oxidation analysis. The remaining part of the liver was immediately snap-frozen in liquid nitrogen and stored at −80 °C until further analysis. 

### 4.2. Plasma Malonyl-CoA

The plasma malonyl-CoA level was determined by the rat malonyl-coenzyme A ELISA kit (MBS2604885, MyBioSource, San Diego, CA, USA).

### 4.3. Plasma Acetylcarnitine, Palmitoylcarnitine, Beta-Hydroxybutyrate, and Acetoacetate

Acetylcarnitine (C2) and palmitoylcarnitine (C16) were analyzed in plasma using HPLC-MS/MS, as described previously [54], with some modifications of the HPLC conditions: the LC system was an Agilent (Waldbronn, Germany) 1200 Series with binary pump, variable volume injector, and a thermostated autosampler. HPLC separation was conducted at 30 °C using a gradient solvent mixture. Mobile phase A was made of 10 mM ammonium acetate and 12 mM HFBA in water, and mobile phase B was made of 10 mM ammonium acetate and 12 mM HFBA in methanol. The gradient was B 0.1 min 20%, flow 0.2 mL/min; B 4 min 20–90%, flow 0.2 mL/min; B 14 min 90%, flow 0.2 mL/min; B 10 min 2%, flow 0.6 mL/min; B 0.1 min 20%, flow 0.2 mL/min. A Phenomenex Luna C8 column (5 μm, 150 × 2 mm) equipped with a Phenomenex C18 precolumn, (4.0 × 2.0 mm) was used. Two microliters of the sample were injected. The ketone bodies β-hydroxybutyrate and acetoacetate were analyzed in plasma by colorimetric assay kits from Cayman Chemical Company (item numbers 700190 and 28352; Ann Arbor, MI, USA).

### 4.4. Plasma and Liver Pyruvate, Lactate, and TCA Cycle Intermediates 

Protein, phospholipids, and particulates were removed from plasma samples with Ostro^TM^ Pass-through 96-well sample-preparation plates (Waters, Milford, MA, USA) according to the supplier’s manual. In brief, the Ostro Plate was placed on a 2 mL collection plate and positioned in the vacuum manifold. An amount of 80 µL sample and 20 µL organic acid internal standard mixture (combination of yeast extract grown with U-^13^C_6_-glucose and prepared in own laboratory and U-^13^C_3_-lactate/pyruvate from Cambridge Isotope Laboratories) were loaded on the Ostro plate wells with 300 µL 1% formic acid in acetonitrile; the sample and solvent were mixed thoroughly 3 times with pipetting. The samples were passed through into a second deep-well plate using a Waters Positive Pressure-96 Processor. The purified plasma samples were dried with a Savant™ SPD131DDA SpeedVac™ (Thermo Fisher Scientific Inc., Waltham, MA, USA) operated for 2.5 h at 4 torr and 60 °C. Samples were kept at −20 °C until further analysis. Prior to derivatization, the samples were reconstituted in ultrapure water. Liver samples (50 ± 2 mg) were homogenized with zirconium oxide beads (0.5 ± 0.01 g, diameter 1.4 mm) in cold 500 µL acetonitrile:H_2_O (55:45) using the Precellys^®^ 24 bead homogenizer equipped with a Cryolys temperature controller (all units from Bertin Technologies SAS, Montigny-le-Bretonneux, France). The settings were 2 × 30 s at 5500 rpm with 5 s pauses. The homogenates were then added to another aliquot of 500 µL acetonitrile:water (55:45), shaken for 20 min at 10 °C on a thermoshaker (Thermal shake lite) (VWR, Radnor, PA, USA), and centrifuged for 10 min at 14,000 rpm (20,000× *g*) at 4 °C (Eppendorf 5810R centrifuge). The supernatant was kept at −20 °C until further analysis.

Organic acids were derivatized as described in [55,56] prior to analysis by liquid chromatography (LC)-MS/MS. Derivatized samples (2 μL) were injected onto a Waters Acquity BEH C18 2.1 × 100 mm column, maintained at 40 °C, and eluted with mobile phases: (A) water added to 0.1% formic acid and (B) methanol. The following gradient (*v*/*v*%) was applied with a flow rate of 0.25 mL/min: 0–0.5 min; 50% B, 0.5–6 min: 50–99% B, 6–7 min: 99% B, 7–7.1 min: 100–50% B, 8 min: end. LC-MS/MS analyses were performed on a Waters Acquity UPLC—Xevo TQ-XS MS/MS system operated in positive electrospray mode. Absolute quantification was performed using the Waters MassLynx V4.1 and based on a dilution series of external standards (Sigma-Aldrich, St. Louis, MO, USA) in combination with the ^13^C-labeled internally added standards. 

### 4.5. Mitochondrial FA Oxidation, Free Fatty Acids, NAD, and Enzyme Activities

Approximately 100 mg of the frozen liver sample was homogenized using a TissueLyzer II (Qiagen, Hilden, Germany) [16]. Palmitoyl-CoA oxidation was measured in the postnuclear fractions as acid-soluble products, as described [16,57]. The amount of protein was measured by the Bio-Rad protein assay kit (Bio-Rad Laboratories, Hercules, CA, USA). Plasma free fatty acids were determined by an enzymatic colorimetric method (WAC0 Nefa C) measured on a COBAS instrument. NAD^+^ and NADH were determined by the NAD/NADH Quantification Colorimetric Kit (ABIN411692, Antibodies-online.com, accessed on 15 August 2019), as per the manufacturer’s protocol. An amount of 20 mg of fresh liver tissue lysate was homogenized with 400 μL of NADH/NAD extraction buffer, and filtered through 10 Kd molecular weight cut-off filters. NADH was detected following the decomposition of NAD by heating to 60 °C for 30 min, and quantified from a NADH standard curve. The enzyme activities of citrate synthase (EC: 2.3.3.1) [58], malonyl-CoA decarboxylase (EC: 4.1.1.9) [59], HMG-CoA synthase (EC: 2.3.3.10) [60], and carnitine palmitoyltransferase (CPT)-2 (EC:2.3.1.21) [61] were measured after isolating the postnuclear fraction of frozen liver samples [17]. Hepatic enzyme activities of alanine transaminase (ALAT), aspartate transaminase (ASAT), and creatine kinase (CK) were measured by a photometry assay.

### 4.6. Hepatic Gene Expression Analysis

Tissue samples (20 mg frozen liver) were homogenized in RNeasy Lysis Buffer from Qiagen (Cat.: 79216, Hilden, Germany) with 1% mercaptoethanol using the Tissuelyser II (Qiagen) for 2 × 2 min at 25 Hz, and total cellular RNA was further purified using the RNeasy mini kit (Qiagen, Hilden, Germany) including DNase digestion. An amount of 500 ng RNA was reverse-transcribed using High-Capacity cDNA Reverse Transcription Kits (Applied Biosystems, Waltham, MA, USA). The qPCR was performed in Sarstedt 384-well Multiply-PCR plates (Sarstedt Inc., Newton, NC, USA) using the ABI Prism 7900HT Sequence Detection system from Applied Biosystems with the software SDS 2.3. Together with the 2× Taqman buffer from Applied Biosystems, the following probes and primers from Applied Biosystems were used to detect mRNA levels of interests. Each probe was run with standard curve using either a representative cDNA sample or cDNA from universal rat reference RNA (URRR, Agilent). Expression levels were normalized to the average of the reference gene large ribosomal protein P0 (Rplp0) (*36b4*, acc.no. M17885), and values relative to the control are shown.

### 4.7. Statistical Analysis

Statistical differences were analyzed using Student’s *t*-test. The *p*-values < 0.05 were considered significant. The results are shown as means ± standard deviation (SD) of 8 rats per group, unless otherwise stated in the figure legends. Correlations were assessed by Spearman’s rho. Results were regarded as statistically significant when *p* < 0.05. The statistical analyses were performed using IBM SPSS Statistics for Windows, Version 25 (IBM Corp., Armonk, NY, USA).

## Figures and Tables

**Figure 1 ijms-24-15536-f001:**
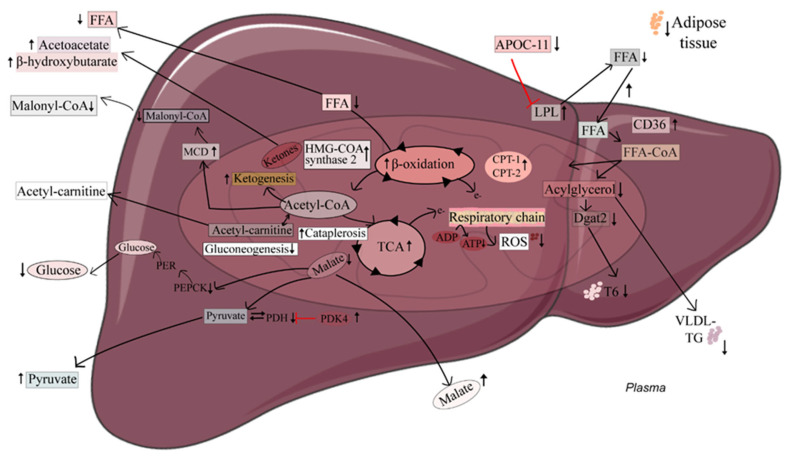
Summary of altered metabolic pathways in association with 3-thia fatty acid (TTA)-induced mitochondrial fatty acid beta-oxidation in the liver. TTA administration to Wistar rats has been shown to increase mitochondrial fatty acid β-oxidation and ketogenesis, while lowering plasma concentrations of free fatty acid, triacylglycerols, and glucose in association with increased hepatic fatty acid uptake and lowered hepatic gluconeogenesis, hepatic ROS generation, and adipose tissue mass. The present study adds insight into the altered activity of the tricarboxylic acid (TCA) cycle in this favorable metabolic context, reflected in, e.g., increased plasma concentrations of pyruvate and malate. Arrows up: increase with TTA administration; Arrows down: decrease with TTA administration.

**Figure 2 ijms-24-15536-f002:**
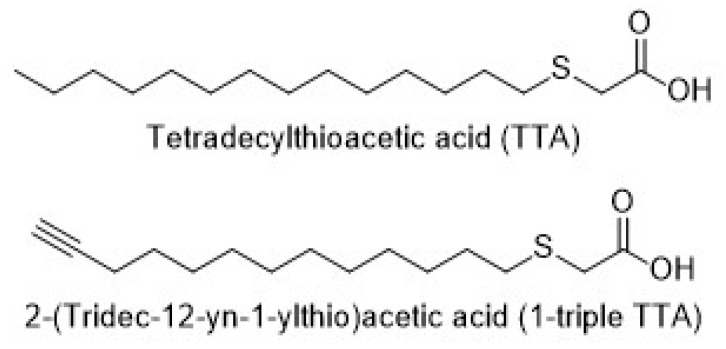
Structure of tetradecylthioacetic acid (TTA) and 2-(tridec-12-yn-1-ylthio)acetic acid.

**Figure 3 ijms-24-15536-f003:**
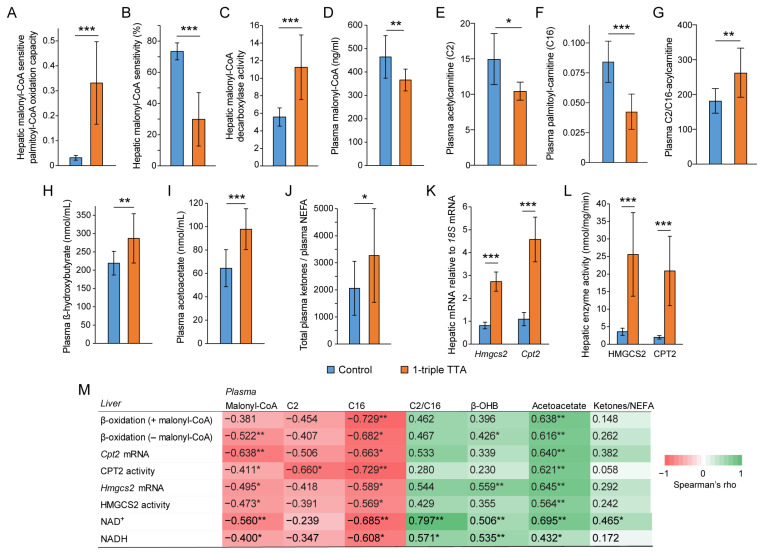
Effects of 3-thia fatty acid treatment on fatty acid beta-oxidation in liver and plasma. Male Wistar rats were treated with the vehicle (control, *n* = 8) or 100 mg/kg body weight 1-tTTA (*n* = 5) for 3 weeks. (**A**,**B**) Malonyl-CoA-sensitive palmitoyl-CoA oxidation capacity (**A**) was measured in liver by radioactivity assay using [1-14C]-palmitoyl-L-carnitine in the absence and present of malonyl-CoA (**B**). (**C**) The concentration of malonyl-CoA was measured in plasma by ELISA. (**D**) The enzyme activity of malonyl-CoA decarboxylase was measured in liver homogenates by a calorimetric assay kit. (**E**,**F**) Plasma carnitines were measured by HPLC-MS/MS, and the ratio of AC2/AC16 (**G**) was calculated for each animal. (**H**–**J**) Ketones were measured by GC-MS/MS, and plasma NEFA was measured by a kit. (**K**) Gene expression was measured by qPCR, and relative expression was calculated and normalized to the reference gene *18S*. (**L**) Hepatic activity of HMGCS2 and CPT2 (nmol/min/mg protein) were measured in postnuclear fractions. (**M**) Correlation matrix showing Spearman’s rho for correlations between hepatic fatty acid β-oxidation or hepatic variables related to β-oxidation (enzyme gene expression and activities, NAD^+^, and NADH) and plasma malonyl-CoA, carnitines, and ketones. Data (**A**–**L**) are presented as mean ± SD. * *p* < 0.05, ** *p* < 0.01, *** *p* < 0.001. Β-OHB, β-hydroxybutyrate; C2, acetylcarnitine; C16, palmitoyl-carnitine; *Cpt2*, carnitine palmitoyl transferase; *Hmgcs2*, 3-hydroxy-3-methylglutaryl-CoA synthase; NEFAs, nonesterified fatty acids; TTA, tetradecylthioacetic acid.

**Figure 4 ijms-24-15536-f004:**
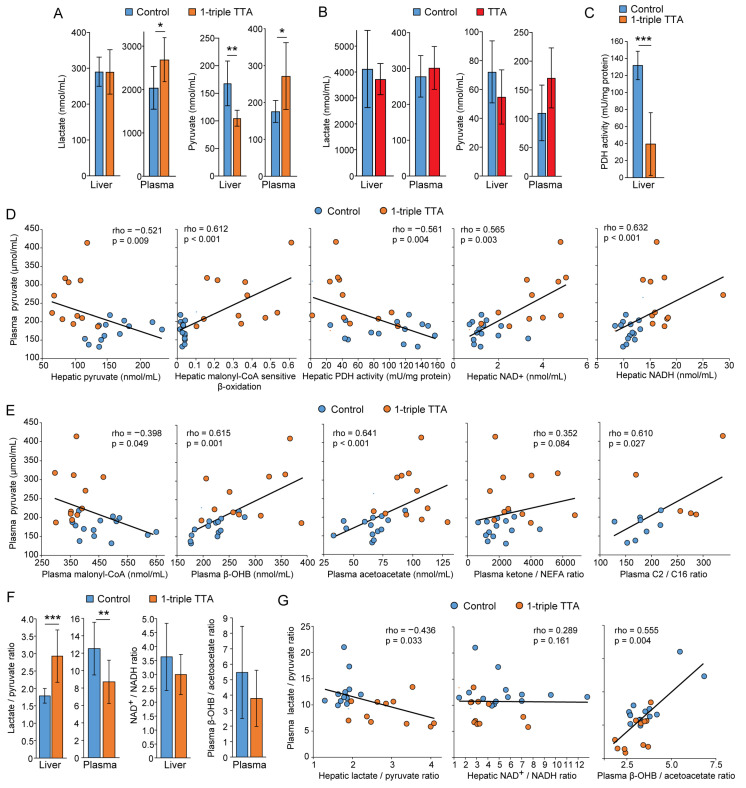
Hepatic and plasma pyruvate reflect the 3-thia fatty acid-induced hepatic FA oxidation. Male Wistar rats were treated with the vehicle (control, *n* = 8) or 100 mg/kg body weight 1-triple TTA (*n* = 5) for 3 weeks (**A**,**C**–**G**), or with the vehicle (control, *n* = 9) or TTA (*n* = 9) for 50 weeks (**B**). (**A**–**E**) Pyruvate and lactate were measured in liver homogenates and plasma by LC-MS/MS. (**F**,**G**) Spearman correlations between hepatic and plasma pyruvate with hepatic mitochondrial fatty acid β-oxidation were calculated. Data are presented as mean ± SD. * *p* < 0.05, ** *p* < 0.01, *** *p* < 0.001. PDH, pyruvate dehydrogenase; TTA, tetradecylthioacetic acid.

**Figure 5 ijms-24-15536-f005:**
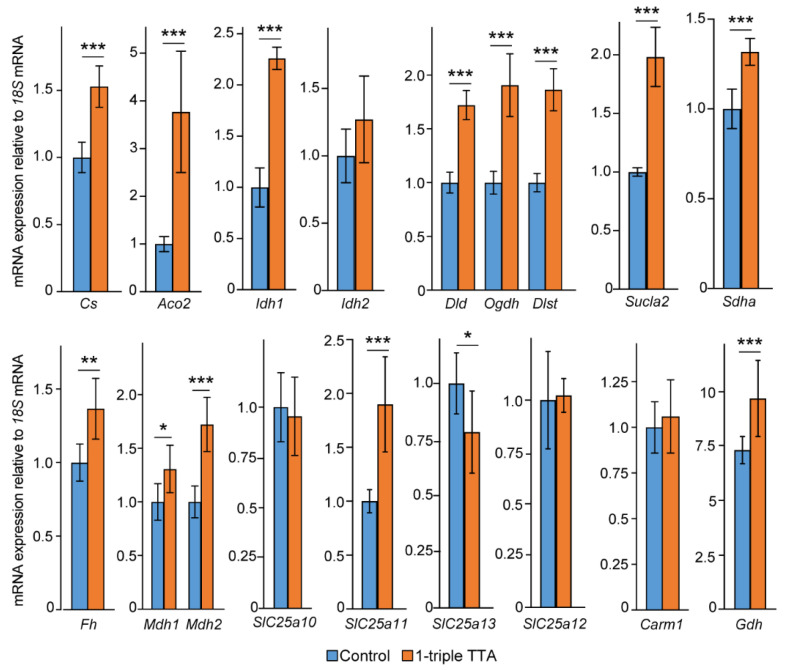
The mRNA expression of TCA-cycle-related enzymes in rats treated with and without 3-thia fatty acids. Male Wistar rats were treated with the vehicle (control, *n* = 8) or 100 mg/kg body weight 1-tTTA (*n* = 5) for 3 weeks. Gene expression was measured by qPCR, and relative expression was calculated and normalized to the reference gene *18S*. Data are presented as mean ± SD. * *p* < 0.05, ** *p* < 0.01, *** *p* < 0.001. *Aco2*, aconitase 2; *Cs*, citrate synthase; *Dld*, dihydrolipoyl dehydrogenase; *Dlst*, dihydrolipoamide S-succinyltransferase; *Fh*, fumarase hydratase; *Ido*, isocitrate dehydrogenase; *Mdh*, malate dehydrogenase; *Ogdh*, 2-oxyglutarate dehydrogenase; *Sdh*, succinate dehydrogenase; *Sucla2*, succinyl-CoA synthase; TTA, tetradecylthioacetic acid.

**Figure 6 ijms-24-15536-f006:**
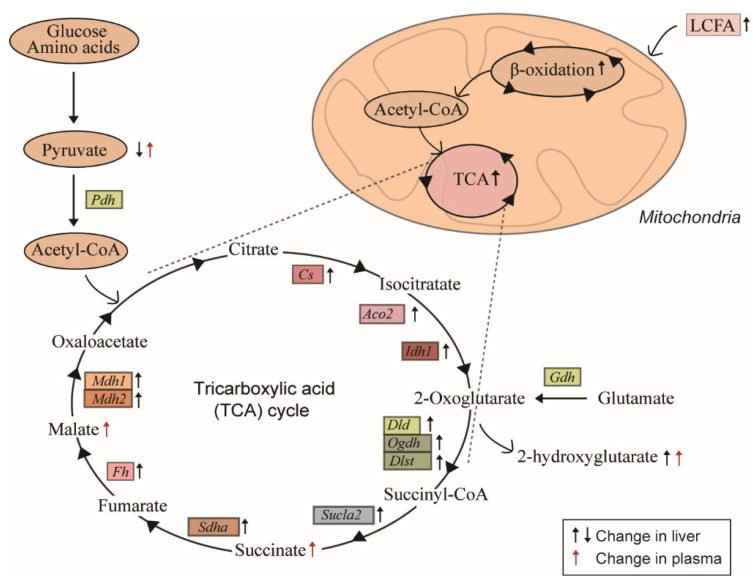
Summary of changes in TCA cycle activity in association with 3-thia fatty acid-induced mitochondrial fatty acid β-oxidation in the liver. The 1-triple TTA treatment of rats for 3 weeks resulted in decreased liver and increased plasma pyruvate concentrations, in association with increased expression of genes encoding enzymes in the TCA cycle. Concentrations of succinate, malate, and 2-hydroxyglutarate were also increased by the treatment in plasma, and 2-hydroxyglutarate concentrations were additionally increased in the liver. LCFA, long-chain fatty acids; TCA, tricarboxylic acid cycle.

**Figure 7 ijms-24-15536-f007:**
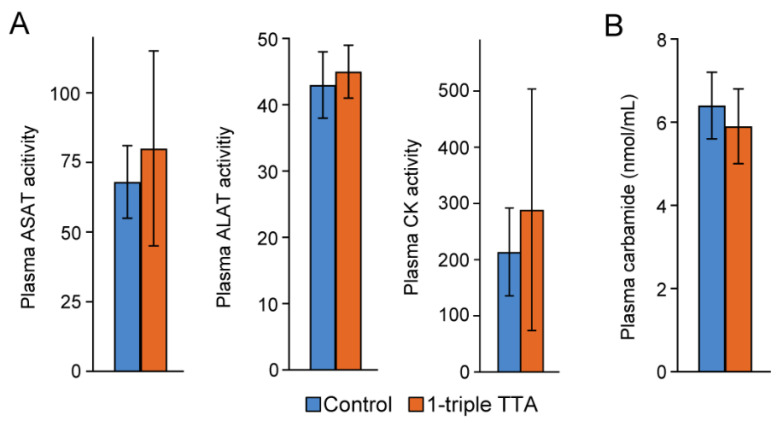
Activity of liver enzymes (**A**) and concentration of carbamide (urea) (**B**) in plasma of rats treated with and without 3-thia fatty acids. Male Wistar rats were treated with the vehicle (control, *n* = 8) or 100 mg/kg body weight 1-triple TTA (*n* = 5) for 3 weeks. Data are presented as mean ± SD. ALAT, alanine transaminase; ASAT, aspartate transaminase; CK, creatine kinase.

**Table 1 ijms-24-15536-t001:** Effects of 3-week 3-thia fatty acid treatment (1-triple TTA, 100 mg/kg body weight) in male Wistar rats on TCA cycle metabolite concentrations in the liver and plasma (control *n* = 8, 1-triple TTA *n* = 5).

Metabolite	Control	1-Triple TTA	*p*-Value	Control	1-Triple TTA	*p*-Value
	Liver (nmol/mg)			Plasma (µM)		
Oxaloacetate	1.76 (1.18)	1.37 (0.71)	0.523	4.36 (2.27)	10.7 (12.3)	0.17
Citrate	0.150 (0.06)	0.103 (0.02)	0.183	234 (36.1)	242 (33.0)	0.70
Isocitrate	N/A	N/A	N/A	10.5 (6.74)	14.6 (6.34)	0.30
2-hydroxyglutarate	0.002 (0.001)	0.009 (0.003)	**<0.001**	0.76 (0.18)	1.14 (0.35)	**0.024**
Succinate	0.984 (0.393)	1.49 (0.743)	0.130	6.98 (1.58)	19.5 (10.8)	**0.007**
Fumarate	0.544 (0.230)	0.433 (0.259)	0.465	2.24 (0.72)	3.0 (1.53)	0.263
Malate	0.098 (0.055)	0.120 (0.06)	0.517	6.51 (2.18)	10.4 (4.08)	**0.043**

Significant *p*-values are highlighted in bold. N/A, not available; TTA, tetradecylthioacetic acid.

## Data Availability

The data are available from the corresponding author on request.

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
