# Peer review of "Changes in Plasma Pyruvate and TCA Cycle Metabolites upon Increased Hepatic Fatty Acid Oxidation and Ketogenesis in Male Wistar Rats"

_ijms, 2023, doi:10.3390/ijms242115536_

Round 1
Reviewer 1 Report
Dankel et al. utilized 3-thia fatty acid tetradecylthioacetic acid (TTA) to enhance hepatic fatty acid beta-oxidation in Wistar rats. Their findings indicated a reduction in hepatic pyruvate levels and concurrent elevations in plasma pyruvate, succinate, malate, and 2-hydroxyglutarate concentrations. The primary objective of this paper was to elucidate alterations in metabolite profiles following TTA treatment.
Here is some mainly issues about this paper:
1. The title can be considered misleading. Upon initial reading, I expected the authors to investigate the broader relationship between plasma metabolites and organ metabolism. However, based on the content and discussions within the paper, the focus centers on modifying hepatic fatty acid oxidation and observing subsequent changes in plasma metabolites. I recommend a more specific title that aligns with the study's primary objective. Additionally, the results section's title, such as the first result "3. -Thia fatty acid-induced alterations in fatty acid oxidation," may require revision, as it inaccurately suggests a broader scope, whereas the authors primarily assess hepatic fatty acid oxidation.
2. The current manuscript contains some fundamental errors. For instance, it appears that Figure 5 does not represent mRNA expression; instead, it illustrates enzyme activity and metabolite concentrations. Furthermore, the legends for Figure 3 lack descriptions of Fig3-L and Fig3-M. I recommend that the authors give meticulous attention to manuscript preparation.
3. The current manuscript lacks effective organization. In the results section, while the authors occasionally specify particular figures (e.g., Fig 3A), they sometimes provide vague references (e.g., "Fig 5-6"). This inconsistency can make it challenging to locate the specific figures to which the authors are referring.
4. Occasionally, the authors assert the existence of data, but it cannot be located within their figures. For instance, in section 260, the authors state, "However, 3-thia fatty acid treatment increased hepatic gene expression of Slc25a11 (Figure 5-6)." However, no such data can be found in Figure 5 or 6, leading to confusion.
In summary, considering the issues identified in the current version of this manuscript, I strongly recommend rejecting its publication.
Author Response
Dankel et al. utilized 3-thia fatty acid tetradecylthioacetic acid (TTA) to enhance hepatic fatty acid beta-oxidation in Wistar rats. Their findings indicated a reduction in hepatic pyruvate levels and concurrent elevations in plasma pyruvate, succinate, malate, and 2-hydroxyglutarate concentrations. The primary objective of this paper was to elucidate alterations in metabolite profiles following TTA treatment.
Here is some mainly issues about this paper:
- The title can be considered misleading. Upon initial reading, I expected the authors to investigate the broader relationship between plasma metabolites and organ metabolism. However, based on the content and discussions within the paper, the focus centers on modifying hepatic fatty acid oxidation and observing subsequent changes in plasma metabolites. I recommend a more specific title that aligns with the study's primary objective. Additionally, the results section's title, such as the first result "3. -Thia fatty acid-induced alterations in fatty acid oxidation," may require revision, as it inaccurately suggests a broader scope, whereas the authors primarily assess hepatic fatty acid oxidation.
Response: We thank the reviewer for the many important comments. We realize that the title and the mentioned subtitle should be more specific. We therefore propose the title “Changes in plasma pyruvate and TCA cycle metabolites upon increased hepatic fatty acid oxidation and ketogenesis in male Wistar rats”, and have added “hepatic” in the subtitle in the first section of the Results. In line with this we have also revised our objective of the study (line 113-114 page 5).
- The current manuscript contains some fundamental errors. For instance, it appears that Figure 5 does not represent mRNA expression; instead, it illustrates enzyme activity and metabolite concentrations. Furthermore, the legends for Figure 3 lack descriptions of Fig3-L and Fig3-M. I recommend that the authors give meticulous attention to manuscript preparation.
Response: Thank you for identifying these errors. We have now included the missing information, double-checked all figure references, and added some more details on methods to avoid confusion.
- The current manuscript lacks effective organization. In the results section, while the authors occasionally specify particular figures (e.g., Fig 3A), they sometimes provide vague references (e.g., "Fig 5-6"). This inconsistency can make it challenging to locate the specific figures to which the authors are referring.
- Occasionally, the authors assert the existence of data, but it cannot be located within their figures. For instance, in section 260, the authors state, "However, 3-thia fatty acid treatment increased hepatic gene expression of Slc25a11 (Figure 5-6)." However, no such data can be found in Figure 5 or 6, leading to confusion.
Response to both 3 and 4: By these comments we see that we accidentally inserted Fig. 7 as both Fig. 5 and 7, and the actual Fig. 5 was missing from the manuscript. We apologize for this, and have now inserted the correct figure and refer only to Fig. 5 to avoid confusion.
In summary, considering the issues identified in the current version of this manuscript, I strongly recommend rejecting its publication.
Response: We regret the unnecessary mistakes, which understandably caused confusion and difficulty in evaluating parts of our paper. During preparation of the revised version, we have carefully checked the figure legends and the references to the figures in the main text.
We hope that the reviewer appreciates the quality of our paper now that these issues are resolved.
Reviewer 2 Report
The manuscript under the code "livers-2548724" with the title: “Plasma pyruvate and TCA cycle metabolites as markers of fatty acid oxidation and ketogenesis in male Wistar rats” aims to determine hepatic and systemic changes in TCA cycle-related metabolites upon selective pharmacologic enhancement of mitochondrial fatty acid β-oxidation in the liver, to elucidate mechanisms and potential markers of hepatic mitochondrial activity. To do that they use Male Wistar rats that were treated with 3-thia fatty acids (e.g.,tetradecylthioacetic acid (TTA)), which target mitochondrial biogenesis, mitochondrial fatty acid β-oxidation and ketogenesis predominantly in the liver. And they measured Hepatic and plasma concentrations of TCA cycle intermediates and anaplerotic substrates, plasma ketones and acylcarnitines, along with associated TCA cycle-related gene expression. And they conclude that Induction of hepatic mitochondrial fatty acid β-oxidation by 3-thia fatty acids lowered hepatic pyruvate while increasing plasma pyruvate as well as succinate, malate and 2-hydroxyglutarate.
General Comments:
The study investigates the impact of 3-thia fatty acid administration on hepatic metabolism in rats. The research is well-structured and provides valuable insights into the effects of this compound on various metabolic parameters. The findings appear to be logically presented, and the methodology seems sound. However, there are a few concerns and suggestions for improvement.
Specific Comments:
Abstract
In the abstract is stated “TCA cycle-related gene expression and enzyme activities (qPCR)”. From my knowledge, I don’t think you can measure enzyme activities with qPCR. Authors should revise the manuscript carefully.
Clarity and Organization:
The paper is generally well-written and organized. However, there are minor typographical errors and sentence structures that need refinement.
It would be helpful to provide a concise overview of the research question and objectives in the introduction section for readers who may not be familiar with the context, beta and omega oxidation…
Methods:
The methods section provides a clear description of the experimental design. However, there is limited information regarding the sample size and the criteria used for selecting rats for the study. This information is essential for assessing the robustness of the study.
Details about the statistical analyses performed should be included to ensure transparency and replicability.
Results and Discussion:
The results section is comprehensive but does not includes all the relevant figures (Figure 5 with the mRNA expression is missing) and tables. And also, there is a lack of interpretation of the results within the results section itself. The authors should provide brief interpretations alongside the presentation of data to aid comprehension.
While the discussion in the manuscript covers a range of findings and implications, there are several gaps and areas where further discussion or exploration could enhance the overall understanding of the research. Here are some of the gaps in the discussion:
-Mechanistic Insights: The discussion could benefit from a deeper exploration of the underlying mechanisms behind the observed changes in pyruvate metabolism and TCA cycle activity. While the study identifies correlations and gene expression changes, more mechanistic insights into how 3-thia fatty acids impact these pathways are needed.
-Gender and Species Differences: The study is limited to male Wistar rats, and it would be valuable to discuss whether these findings can be generalized to other genders or species. Addressing potential gender differences or variations in different animal models would strengthen the discussion.
-Long-Term Effects: The manuscript briefly mentions the potential long-term effects of 3-thia fatty acid administration but does not delve into these consequences. Discussing the implications of sustained changes in pyruvate and TCA cycle metabolites over time would provide a more comprehensive perspective.
-Clinical Relevance: While the study identifies potential plasma markers associated with enhanced fatty acid β-oxidation, it would be beneficial to discuss the clinical relevance of these findings. How might these markers be applied or investigated in the context of human metabolic diseases or therapies?
-Mitochondrial Redox State: The discussion mentions the impact of the study on mitochondrial redox state briefly. Expanding on how these changes in redox state might relate to mitochondrial function and cellular health would be informative.
-Limitations: The manuscript acknowledges several limitations, such as the snapshot nature of the analysis and the need for future enzyme activity and flux studies. Further discussion of how these limitations might affect the interpretation of results and the direction of future research would be valuable.
-Future Research Directions: Concluding the discussion with a section on future research directions based on the current findings would provide a roadmap for researchers interested in building upon this work.
Certainly, here are some experiments that could be conducted to address gaps in the manuscript and further investigate the findings:
-Measurement of Enzyme Activities: To better understand the mechanisms behind the altered pyruvate metabolism and TCA cycle activity, you can measure the activities of key enzymes involved in these processes. Specifically, measure the activities of pyruvate carboxylase (PC), pyruvate dehydrogenase (PDH), and other relevant enzymes in liver tissues from the rats treated with 3-thia fatty acids.
-Metabolomic Profiling: Perform a comprehensive metabolomic profiling analysis on liver tissues and plasma samples to identify changes in metabolite concentrations beyond those mentioned in the study. This can provide a more detailed picture of the metabolic alterations induced by 3-thia fatty acids.
-Tissue-Specific Redox State Analysis: Investigate the redox state not only in the liver but also in other metabolically active tissues, such as skeletal muscle and adipose tissue. This will help determine if the observed changes in pyruvate and TCA cycle metabolites are liver-specific or systemic.
-Tracer Studies: Use stable isotope tracers to trace the fate of specific metabolic substrates (e.g., glucose, fatty acids, and pyruvate) in different metabolic pathways. This can provide insights into the flux of metabolites and help elucidate the origin of changes in plasma and tissue metabolite concentrations.
-Inhibition of Key Enzymes: To establish causality, conduct experiments where key enzymes involved in pyruvate metabolism and the TCA cycle (e.g., PC, PDH) are inhibited pharmacologically or genetically. This will help confirm whether changes in enzyme activities directly contribute to the observed alterations in metabolite profiles.
-Mitochondrial Functional Assays: Assess mitochondrial function directly by measuring parameters such as mitochondrial respiration, ATP production, and mitochondrial membrane potential in liver mitochondria isolated from treated rats. This can link changes in mitochondrial function to altered metabolite profiles.
-ROS and Antioxidant Status: Measure reactive oxygen species (ROS) levels and antioxidant enzyme activities in liver tissues to determine whether changes in redox state play a role in the observed metabolic alterations.
These experiments can help fill gaps in the manuscript, provide a more comprehensive understanding of the metabolic effects of 3-thia fatty acids, and further validate the findings.
Figures and Tables:
The figures and tables are generally clear and relevant. But as i mention before there are some figures missing.
References:
The references are appropriately cited, but some of them are outdated. The authors should consider including more recent references to ensure the study aligns with the current state of the field.
Overall Assessment:
This research paper presents valuable insights into the effects of 3-thia fatty acid administration on hepatic metabolism in rats. With some minor revisions to improve clarity, transparency, and interpretation of results, this study has the potential to add info to the field. Addressing some of the gaps in the discussion would not only strengthen the overall conclusions of the study but also guide future research in this area. I recommend address some of the above concerns.
The paper is generally well-written and organized. However, there are minor typographical errors and sentence structures that need refinement. The authors should consider a thorough proofreading and editing process.
Author Response
The manuscript under the code "livers-2548724" with the title: “Plasma pyruvate and TCA cycle metabolites as markers of fatty acid oxidation and ketogenesis in male Wistar rats” aims to determine hepatic and systemic changes in TCA cycle-related metabolites upon selective pharmacologic enhancement of mitochondrial fatty acid β-oxidation in the liver, to elucidate mechanisms and potential markers of hepatic mitochondrial activity. To do that they use Male Wistar rats that were treated with 3-thia fatty acids (e.g.,tetradecylthioacetic acid (TTA)), which target mitochondrial biogenesis, mitochondrial fatty acid β-oxidation and ketogenesis predominantly in the liver. And they measured Hepatic and plasma concentrations of TCA cycle intermediates and anaplerotic substrates, plasma ketones and acylcarnitines, along with associated TCA cycle-related gene expression. And they conclude that Induction of hepatic mitochondrial fatty acid β-oxidation by 3-thia fatty acids lowered hepatic pyruvate while increasing plasma pyruvate as well as succinate, malate and 2-hydroxyglutarate.
General Comments:
The study investigates the impact of 3-thia fatty acid administration on hepatic metabolism in rats. The research is well-structured and provides valuable insights into the effects of this compound on various metabolic parameters. The findings appear to be logically presented, and the methodology seems sound. However, there are a few concerns and suggestions for improvement.
Specific Comments:
Abstract
In the abstract is stated “TCA cycle-related gene expression and enzyme activities (qPCR)”. From my knowledge, I don’t think you can measure enzyme activities with qPCR. Authors should revise the manuscript carefully.
Response: Thank you for pointing this out, the parenthesis denoting qPCR has now been positioned right after “gene expression”.
Clarity and Organization:
The paper is generally well-written and organized. However, there are minor typographical errors and sentence structures that need refinement.
It would be helpful to provide a concise overview of the research question and objectives in the introduction section for readers who may not be familiar with the context, beta and omega oxidation…
Response: Thank you for the suggestion to further improve the introduction. To provide more general context, we have now added the following sentence in the beginning: “Cells obtain the chemical energy stored in fatty acids (FAs) by sequential degradation of the fatty acids, preferentially via β-oxidation in the peroxisomes (very long-chain FAs) and mitochondria (long-, medium- and short-chain FAs), or via α-oxidation (peroxisomes) or ω-oxidation (endoplasmic reticulum) when the carbon-3 of FAs has a methyl- or other functional groups.” Moreover, we have attempted to clarify the objective by reformulating parts of the text towards the end of the introduction.
Methods:
The methods section provides a clear description of the experimental design. However, there is limited information regarding the sample size and the criteria used for selecting rats for the study. This information is essential for assessing the robustness of the study.
Details about the statistical analyses performed should be included to ensure transparency and replicability.
Response: We have added the following to the methods section that introduces the rat experiment: “Due to the potent metabolic effects of 3-thia FAs, previous experiments found marked differences in metabolic phenotypes with as few as 3 rats per group, supporting robust power in the present experiment." Regarding statistical analyses, we have now specified that results are considered statistically significant when p<0.05, and indicated the p-value thresholds in the figure legends where relevant.
Results and Discussion:
The results section is comprehensive but does not includes all the relevant figures (Figure 5 with the mRNA expression is missing) and tables. And also, there is a lack of interpretation of the results within the results section itself. The authors should provide brief interpretations alongside the presentation of data to aid comprehension.
Response: We apologize for mistakenly inserting a duplicate of Fig. 7 where Fig. 5 (showing the gene expression) was supposed to be. This has now been corrected. We have also added sentences in the Results to help with the interpretation.
While the discussion in the manuscript covers a range of findings and implications, there are several gaps and areas where further discussion or exploration could enhance the overall understanding of the research. Here are some of the gaps in the discussion:
-Mechanistic Insights: The discussion could benefit from a deeper exploration of the underlying mechanisms behind the observed changes in pyruvate metabolism and TCA cycle activity. While the study identifies correlations and gene expression changes, more mechanistic insights into how 3-thia fatty acids impact these pathways are needed.
Response: We thank the reviewer for this relevant request for a more detailed discussion of possible mechanisms, and have added a new paragraph towards the end of the Discussion.
-Gender and Species Differences: The study is limited to male Wistar rats, and it would be valuable to discuss whether these findings can be generalized to other genders or species. Addressing potential gender differences or variations in different animal models would strengthen the discussion.
Response: We appreciate this point, which we now address in the revised limitations paragraph of the Discussion.
-Long-Term Effects: The manuscript briefly mentions the potential long-term effects of 3-thia fatty acid administration but does not delve into these consequences. Discussing the implications of sustained changes in pyruvate and TCA cycle metabolites over time would provide a more comprehensive perspective.
Response: We have expanded the discussion of longer-term effects of 3-thia fatty acid administration, in the same new paragraph that discusses mechanisms.
-Clinical Relevance: While the study identifies potential plasma markers associated with enhanced fatty acid β-oxidation, it would be beneficial to discuss the clinical relevance of these findings. How might these markers be applied or investigated in the context of human metabolic diseases or therapies?
Response: We have added the following sentence at the end of the conclusions paragraph to provide an example of the potential clinical relevance of our findings: “For example, increased serum pyruvate was recently reported as a prognostic marker for early detection of patients at risk of critical COVID-19, and our data suggest that this risk might involve altered energy metabolism in the liver.”
-Mitochondrial Redox State: The discussion mentions the impact of the study on mitochondrial redox state briefly. Expanding on how these changes in redox state might relate to mitochondrial function and cellular health would be informative.
Response: We have attempted to clarify some of the discussion in the relevant paragraph of the Discussion, while trying to avoid excessive expansion of an already long discussion (especially with the new paragraph on mechanisms).
-Limitations: The manuscript acknowledges several limitations, such as the snapshot nature of the analysis and the need for future enzyme activity and flux studies. Further discussion of how these limitations might affect the interpretation of results and the direction of future research would be valuable.
-Future Research Directions: Concluding the discussion with a section on future research directions based on the current findings would provide a roadmap for researchers interested in building upon this work.
Response (addressing the last two points): We have added the following sentence to the limitations paragraph: “For example, use of stable isotope tracers would allow tracing the fate of specific metabolic substrates (e.g., of glucose, fatty acids, and pyruvate) to more precisely understand metabolite fluxes and the origins of plasma and tissue metabolite changes 48,49. Also, the impact of pharmacologic enzyme inhibition on liver and plasma metabolites could be investigated, and redox states and metabolites could be measured in other tissues that may contribute to plasma metabolite concentrations such as skeletal muscle, adipose tissue and the heart.”
Certainly, here are some experiments that could be conducted to address gaps in the manuscript and further investigate the findings:
-Measurement of Enzyme Activities: To better understand the mechanisms behind the altered pyruvate metabolism and TCA cycle activity, you can measure the activities of key enzymes involved in these processes. Specifically, measure the activities of pyruvate carboxylase (PC), pyruvate dehydrogenase (PDH), and other relevant enzymes in liver tissues from the rats treated with 3-thia fatty acids.
-Metabolomic Profiling: Perform a comprehensive metabolomic profiling analysis on liver tissues and plasma samples to identify changes in metabolite concentrations beyond those mentioned in the study. This can provide a more detailed picture of the metabolic alterations induced by 3-thia fatty acids.
-Tissue-Specific Redox State Analysis: Investigate the redox state not only in the liver but also in other metabolically active tissues, such as skeletal muscle and adipose tissue. This will help determine if the observed changes in pyruvate and TCA cycle metabolites are liver-specific or systemic.
-Tracer Studies: Use stable isotope tracers to trace the fate of specific metabolic substrates (e.g., glucose, fatty acids, and pyruvate) in different metabolic pathways. This can provide insights into the flux of metabolites and help elucidate the origin of changes in plasma and tissue metabolite concentrations.
-Inhibition of Key Enzymes: To establish causality, conduct experiments where key enzymes involved in pyruvate metabolism and the TCA cycle (e.g., PC, PDH) are inhibited pharmacologically or genetically. This will help confirm whether changes in enzyme activities directly contribute to the observed alterations in metabolite profiles.
-Mitochondrial Functional Assays: Assess mitochondrial function directly by measuring parameters such as mitochondrial respiration, ATP production, and mitochondrial membrane potential in liver mitochondria isolated from treated rats. This can link changes in mitochondrial function to altered metabolite profiles.
-ROS and Antioxidant Status: Measure reactive oxygen species (ROS) levels and antioxidant enzyme activities in liver tissues to determine whether changes in redox state play a role in the observed metabolic alterations.
These experiments can help fill gaps in the manuscript, provide a more comprehensive understanding of the metabolic effects of 3-thia fatty acids, and further validate the findings.
Response: We greatly appreciate the many insightful suggestions for improvements and possibilities for further research. Please note that some of the data are already presented (PDH activity, Fig. 4C) and that several data were published previously, which we now refer to (e.g., several other metabolites, new ref 45; antioxidant properties, new ref 47). Beyond this we have added many of the above-mentioned suggestions for further research in the final paragraph of the Discussion.
Figures and Tables:
The figures and tables are generally clear and relevant. But as i mention before there are some figures missing.
References:
The references are appropriately cited, but some of them are outdated. The authors should consider including more recent references to ensure the study aligns with the current state of the field.
Response: As noted above we have inserted the correct figure showing the gene expression data, and again apologize for this mistake. Thank you for the suggestion regarding references, we have now added several new and more recent references.
Overall Assessment:
This research paper presents valuable insights into the effects of 3-thia fatty acid administration on hepatic metabolism in rats. With some minor revisions to improve clarity, transparency, and interpretation of results, this study has the potential to add info to the field. Addressing some of the gaps in the discussion would not only strengthen the overall conclusions of the study but also guide future research in this area. I recommend address some of the above concerns.
Response: We thank the reviewer again for the many good suggestions, and hope our responses and improvements of the manuscript are found to be satisfactory.
Round 2
Reviewer 1 Report
I believe this version of the manuscript is much improved, and I don't have any further complaints.